



# Enhancement of secondary aerosol formation by reduced anthropogenic emissions during Spring Festival 2019 and enlightenment for regional PM2.5 control in Beijing

Yuying Wang[1], Zhanqing Li[2], Qiuyan Wang[1], Xiaoai Jin[3], Peng Yan[4], Maureen Cribb[2], Yanan Li[4], Cheng Yuan[1], Hao Wu[3], Tong Wu[3], Rongmin Ren[3], Zhaoxin Cai[3]

[1]Key Laboratory for Aerosol-Cloud-Precipitation of China Meteorological Administration, School of Atmospheric Physics, Nanjing University of Information Science & Technology, Nanjing 210044, China

[2]Earth System Science Interdisciplinary Center, Department of Atmospheric and Oceanic Science, University of Maryland, College Park, MD, USA

[3]State Key Laboratory of Remote Sensing Science, College of Global Change and Earth System Science, Beijing Normal University, Beijing 100875, China

[4]CMA Meteorological Observation Center, Centre for Atmosphere Watch and Services, Beijing 100081, China

Correspondence to: Yuying Wang (yuyingwang@nuist.edu.cn)





**Abstract**
A comprehensive field experiment measuring aerosol chemical and physical
properties at a suburban site in Beijing around the 2019 Spring Festival was carried out
to investigate the impact of reduced anthropogenic emissions on aerosol formation.
Sharply reduced sulfur dioxide ($SO_2$) and nitrogen dioxide ($NO_2$) concentrations during
the festival holiday resulted in an unexpected increase in the surface ozone ($O_3$)
concentration, leading to enhancement of the atmospheric oxidation capacity.
Simultaneously, the reduced anthropogenic emissions resulted in massive decreases in
particle number concentration at all sizes and the mass concentrations of organics and
black carbon. However, the mass concentrations of inorganics (especially sulfate)
decreased weakly. Detailed analyses of the sulfur oxidation ratio and the nitrogen
oxidation ratio suggest that sulfate formation during the holiday could be promoted by
enhanced nocturnal aqueous-phase chemical reactions between $SO_2$ and $O_3$ under
moderate relative humidity (RH) conditions (40 % < RH < 80 %). Daytime
photochemical reactions in winter in Beijing mainly controlled nitrate formation, which
was enhanced a little during the holiday. A regional analysis of air pollution patterns
shows that the enhanced formation of secondary aerosols occurred throughout the entire
Beijing-Tian-Hebei (BTH) region during the holiday, partly offsetting the decrease in
particle matter with an aerodynamic diameter less than 2.5 μm. Our results highlight
the necessary control of $O_3$ formation to reduce secondary pollution in winter. The
emission control of volatile organic compounds (VOCs) may be more suitable than the
emission control of $NO_x$ to reduce $O_3$ because VOCs under current emission conditions



likely control the formation of $O_3$ in winter in the BTH region.

## 1.  Introduction

Aerosols consist of liquid and solid particles, and their mixture suspended in the

atmosphere. The massive increase in aerosol particles caused by human activities (e.g.,
traffic, industrial production, and construction work) in urban areas can deteriorate air
quality to the point of having a detrimental impact on human health (e.g., Chow et al.,
2006; Matus et al., 2012; Gao et al., 2017; Zhong et al., 2018; An et al., 2019).
Moreover, aerosols can change atmospheric optical and hygroscopic properties, altering
the transfer of solar radiation and the development of clouds, thereby changing weather
and climate in both aerosol source regions and their downstream areas (e.g., Altaratz et
al., 2014; R. Zhang et al., 2015; Z. Li et al., 2016, 2019; Y. Wang et al., 2018, 2019b;
Jin et al., 2020).

With the rapid economic development and urbanization in recent decades in China,

the scales of many cities have expanded quickly along with sharply increased
populations in urban areas, especially in the three most economically developed regions
(the Beijing-Tianjin-Hebei (BTH) metropolitan region, the Yangtze River Delta, and
the Pearl River Delta). As a result, air pollution has become a severe problem in these
megacity regions (e.g., Chan and Yao, 2008; Han et al., 2014; Zhong et al., 2018). On
some heavy haze days, the mass concentration of particulate matter with an
aerodynamic diameter of less than 2.5 μm ($PM_{2.5}$) dramatically increased from tens to
hundreds of micrograms per cubic meter in several hours (Guo et al., 2014; Sun et al.,


2016a).

Over the past a few years, many emission control measures have been taken in

China to mitigate air pollution. As a response, the mass concentration of $PM_{2.5}$ has
decreased in most cities in China since 2013, especially in the BTH region (Q. Zhang
et al., 2019; Vu et al., 2019; Zhai et al., 2019). Organics and black carbon (BC)
concentrations largely decreased during these years thanks to the reduction in coal
combustion and biomass burning (H. Li et al., 2019a; Xu et al., 2019). Simultaneously,
the mass concentrations of inorganics (mainly sulfate, nitrate, and ammonium) also
decreased due to the reduction in their gaseous precursors (especially sulfur dioxide, or
$SO_2$). However, the mass fraction of inorganics increased by more than 10 % during
these years (H. Li et al., 2019a; Y. Wang et al., 2019a), implying the enhancement of
secondary aerosol formation, which partly counteracted the decrease in $PM_{2.5}$.
Therefore, elaborating the secondary aerosol formation mechanism under current
emission conditions is important for taking more proper measures to control $PM_{2.5}$ in
the future.

Some studies have argued that controlling emissions of nitrogen oxides ($NO_x$) is

important because nitrate in $PM_{2.5}$ has had the weakest decrease relative to other
chemical species over the past several years (Q. Zhang et al., 2019; F. Zhang et al.,
2020). The transformation of $NO_x$ to nitrate is closely related to atmospheric oxidation
processes (Seinfeld and Pandis, 2016). Surface ozone ($O_3$) is an important secondary
gaseous pollutant and oxidizing agent in the atmosphere. Recent studies have found
that a reduction in $PM_{2.5}$ resulted in an increase in the $O_3$ volume mixing ratio ($[O_3]$) at



a rate of 3.3 ppbv per annum during the summer of the past few years in the BTH region
(K. Li et al., 2019, 2020). The increased $[O_3]$ can enhance the atmospheric oxidation
capacity, thereby promoting the formation of secondary aerosols in summer (T. Wang
et al., 2017). However, less emphasis has been placed on the variation in $[O_3]$ in winter.
The formation of $O_3$ and its effect on secondary aerosol formation in a cold environment
is thus unclear.
Some special events held in China have provided unique opportunities to
investigate the impact of human activities on air quality by taking advantage of unusual
changes associated with short-term, drastic measures implemented by the Chinese
government to reduce anthropogenic emissions, such as the 2008 Summer Olympic
Games (T. Wang et al., 2010; Guo et al., 2013), the 2014 Asia-Pacific Economic
Cooperation (Sun et al., 2016b), the 2015 China Victory Day parade (Y. Wang et al.,
2017; Zhao et al., 2017), and the 2016 G20 Summit (H. Li et al., 2019b). The annual
Spring Festival holiday is also a special occasion when the vast majority of the
population stops working for 2 to 4 weeks (Tan et al., 2009; Y. Zhang et al., 2016; C.
Wang et al., 2017). Investigating the impact of changes in anthropogenic emissions on
gaseous pollutants and aerosol formation during these special occasions may provide
useful guidance on more scientifically sound measures to take to control $PM_{2.5}$.
A comprehensive aerosol field experiment at a suburban site near the 5th Ring
Road in the Daxing District of Beijing was carried out for more than two years,
including the 2019 Spring Festival. Beijing was one of the three top cities in China with
the largest migrating population during the 2019 Spring Festival holiday



(https://cloud.tencent.com/developer/news/393324). In addition, fireworks were
prohibited throughout the Beijing metropolitan region within the 6th Ring of the Beijing
Beltway. The intensity of anthropogenic emissions was thus much weaker than usual
during this holiday. Our measurements made around this period of the field campaign
are thus ideal for investigating the impact of reduced anthropogenic emissions on
surface $O_3$ and aerosol formation.
This paper is structured as follows. Section 2 describes the experiment site and the
measurement data used in this study. Section 3 presents the results and discussion,
mainly concerning the impact of reduced anthropogenic emissions during the holiday
on the variations in trace gases and aerosol chemical species in Beijing and the BTH
region. Section 4 presents the conclusions and their implications.

### 101  2.  Experiment site and measurement data

A comprehensive field experiment measuring aerosol physical and chemical
properties was conducted from August 2017 to October 2019 at a suburban site in
southern Beijing (Fig. 1). Note that this study only employs measurements made
around the 2019 Spring Festival from 16 January to 17 February. This site (39.81ºN,
116.48ºE) is the test center for meteorological instruments constructed by the China
Meteorological Administration (CMA). It is surrounded by Beijing's 5th Ring Road,
industrial parks, and residential communities (Fig. S1). Aerosol chemical and physical
properties in this area are thus mainly anthropogenic, varying considerably around the
time of the festival in response to the full cycle of industrial activities as the majority





of people stopped and resumed working. This provides an opportunity to investigate
the impact of reduced anthropogenic emissions on surface $O_3$ and aerosol formation
processes in winter.

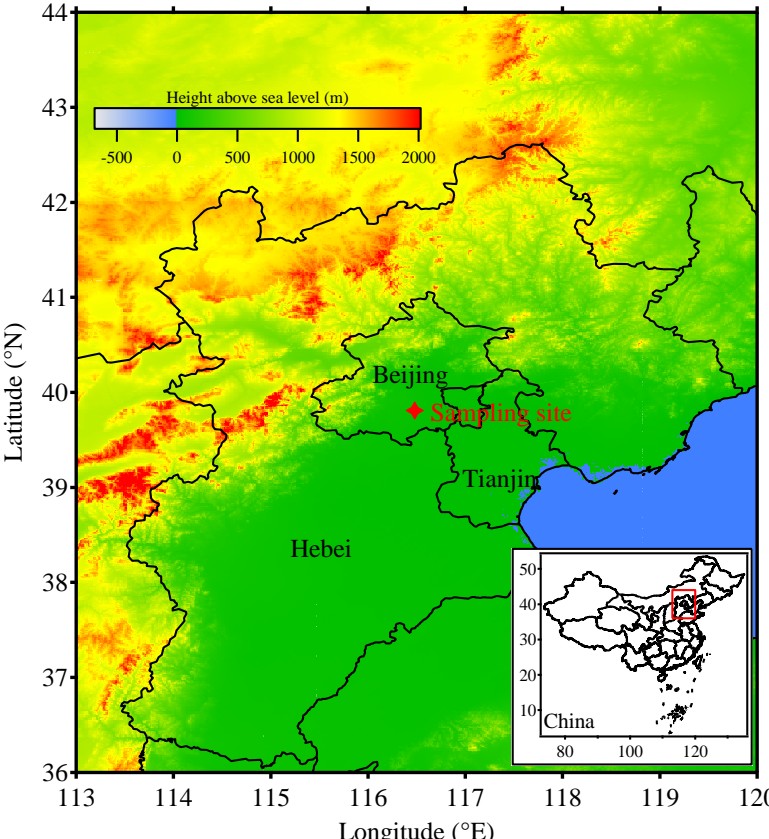


**Figure 1**. Map showing the Beijing-Tianjin-Hebei region in China and the location of
the experiment site. The colored background shows the terrain height (unit: m above
sea level).

Table 1 lists the instruments used in this campaign. A scanning mobility particle

sizer (SMPS) and an aerodynamic particle sizer (APS) measured the aerosol particle




number size distribution (PNSD) from 10 nm to 20 μm. The SMPS consists of a
differential mobility analyzer (model 3081, TSI Inc.) and a condensation particle
counter (model 3772, TSI Inc.). The aerodynamic diameter measured by the APS can
be converted to the Stokes diameter through division by the square root of the aerosol
density. The aerosol density in this study was calculated following the method of Zhao
et al. (2017), using measured aerosol chemical composition information. An aerosol
chemical speciation monitor (ACSM) equipped with a $PM_{2.5}$ lens system, a capture
vaporizer, and a quadrupole mass spectrometer was used to measure mass
concentrations of non-refractory aerosol chemical species in $PM_{2.5}$, including organics
(Org), nitrate ($NO_3^-$), sulfate ($SO_4^{2-}$), ammonium ($NH_4^+$), and chlorine (Chl) (Peck et al.,
2016; Xu et al., 2017; Y. Zhang et al., 2017). A seven-wavelength aethalometer (model
AE-33, Magee Scientific Corp.) with a $PM_{2.5}$ cyclone in the sample inlet was used to
retrieve the mass concentration of BC.

In addition to the above aerosol measurements, meteorological parameters were

observed by the CMA at the experiment site. The Chinese Ministry of Ecology and
Environment network and Beijing Municipal Environmental Monitoring Center
(http://106.37.208.233:20035/ and http://www.bjmemc.com.cn/) provided $PM_{2.5}$ and
trace gas (sulfur dioxide ($SO_2$), nitrogen dioxide ($NO_2$), carbon monoxide (CO), and
$O_3$) measurements made in different locations of the BTH region. Yizhuang in Beijing
is the nearest station to the experiment site (about 3.0 km to the southeast, Fig. S1). The
total mass concentrations of measured non-refractory aerosol chemical species and BC
mass concentrations in $PM_{2.5}$ show good consistency with the $PM_{2.5}$ mass



concentrations obtained from the Yizhuang station (Fig. S2).

**Table 1**. Aerosol instruments used in this campaign and their observed parameters and
manufacturer information.

| Instrument | Measured Parameters | Manufacturer | Model | Time Resolution |
|---|---|---|---|---|
| SMPS | Particle number size distribution (10–550 nm) | TSI | 3938 | 5 min |
| APS | Particle number size distribution (0.5–20 μm) | TSI | 3321 | 5 min |
| ACSM | Mass concentrations of non-refractory aerosol chemical species in $PM_{2.5}$ | Aerodyne | Q-ACSM | 15 min |
| Aethalometer | Mass concentration of black carbon | Magee | AE-33 | 5 min |


**3.    Results and Discussion**
**3.1.    Basic meteorological and environmental characteristics**

While the official Spring Festival holiday was from 4 February to 10 February

2019, many people left before 3 February and came back after the Lantern Festival (19
February). In this study, we regarded the days from 16 January to 2 February as the
polluted (POL) period with high anthropogenic emissions, more representative of
ordinary conditions, and the days from 3 February to 17 February as the background
(BG) period with low anthropogenic emissions. Figure 2 shows the time series of
meteorological parameters, trace gas volume mixing ratios, and aerosol properties
during the two periods.



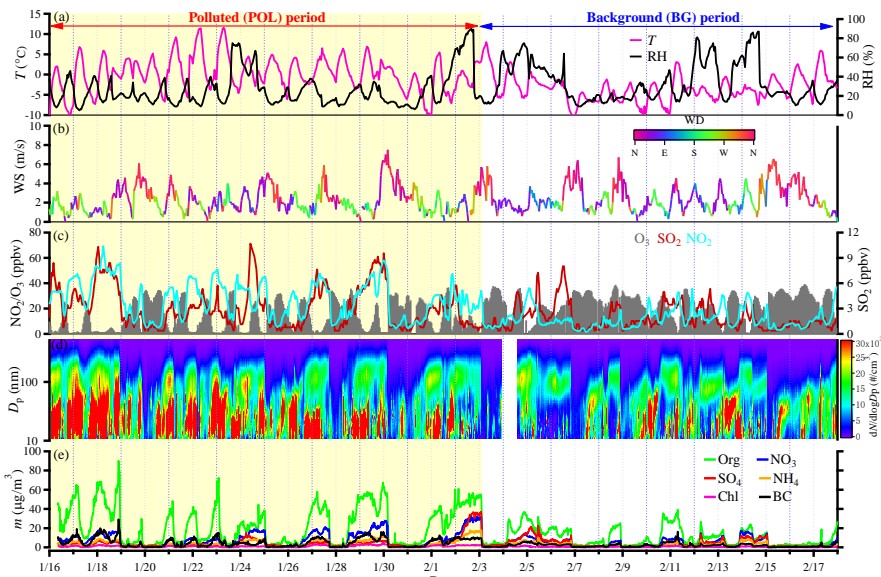

**Figure 2.** Time series of (**a**) ambient temperature ($T$) and relative humidity (RH), (**b**) wind direction (WD) and speed (WS), (**c**) volume mixing ratios of trace gases ($O_3$, $SO_2$, and $NO_2$), (**d**) the aerosol particle number size distribution measured by the SMPS, and (**e**) mass concentrations of aerosol chemical species in $PM_{2.5}$ measured by the ACSM and the AE-33. The trace gas information was from the Yizhuang station, and the others were observed at the experiment site in Beijing (16 January to 17 February 2019).

Ambient temperature ($T$) and relative humidity (RH) have clear diurnal cycles (Fig. 2a). The average $T$ and RH during the BG period were slightly lower (-3.3±3.4 versus 0.2±4.2°C) and higher (33.2±20.1 versus 25.8±17.6 %) than those during the POL period, respectively. This was caused by several short-term light snowfall events that occurred on 6, 12, and 14 February during the BG period. Figures 2b and S3 display similar wind patterns during the POL and BG periods, i.e., wind patterns that changed



periodically. The prevailing, strong northerly winds during the two periods were
beneficial to dispersing pollutants in Beijing (Sun et al., 2016b; Y. Wang et al., 2017),
and thus no heavy haze episodes occurred during these periods. Overall, the
meteorological parameters were similar during the POL and BG periods.

Figure 2c depicts that the volume mixing ratios of $SO_2$ and $NO_2$ ([$SO_2$] and [$SO_2$])

during the BG period were lower than those during the POL period, suggesting less
gaseous pollutants from anthropogenic emissions during the BG period. In addition,
[$O_3$] remained at a high level for several days during the BG period but not during the
POL period. The average [$O_3$] increased by 77.4 % during the BG period compared
with the POL period (46.2±18.9 versus 26.1±22.2 ppbv). The percent change in [$O_3$]
due to the "holiday effect" during this field campaign is much higher than that reported
in other regions of China (K. Huang et al., 2012; C. Wang et al., 2017; S. Wang et al.,

2019).

Many bursts of fine particles (Fig. 2d) occurring mainly during rush hours or at

night were observed during the POL period. This is likely related to the substantial
increases in gasoline or diesel vehicles on two nearby roads at these times. Zhu et al.
(2017) found that efficient nucleation and partitioning of gaseous species from on-road
vehicles can promote new particle formation in the wintertime. However, this
phenomenon occurred much less frequently during the BG period, likely because of the
massive reduction in on-road vehicles. The few short-term bursts of fine particles
during the BG period occurred during the daytime, presumably because of enhanced
nucleation by photochemical processes.





The aerosol chemical species in $PM_{2.5}$ also differed during the POL and BG periods
(Fig. 2e). During the POL period, the mass concentrations of aerosol chemical species
readily accumulated, especially the organics ($m_{org}$) with rapid increases at night. The
mass concentration of BC ($m_{BC}$) also clearly increased, likely associated with an
increase in heavy-duty diesel vehicles and a decrease in the nocturnal planetary
boundary layer at night (Y. Wang et al., 2017; Zhao et al., 2017; Z. Li et al., 2017).
However, the increases in $m_{org}$ and $m_{BC}$ during the BG period were not as strong as
those during the POL period. The mass concentration of nitrate ($m_{NO3}$) largely decreased
during the BG period, while there was a weak variation in the mass concentration of
sulfate ($m_{SO4}$).
In summary, distinct differences existed in all observed trace gases and aerosol
chemical and physical parameters during the POL and BG periods. However, the
meteorological parameters (wind direction and speed, ambient temperature, and RH)
and weather regimes were similar during these two periods. This helps to single out the
impact of reduced anthropogenic emissions on trace gases and aerosol formation
processes.

**3.2.   Impact of reduced anthropogenic emissions on aerosol formation processes**
The average $PM_{2.5}$ mass concentrations were 46.3 and 22.5 μg/m$^3$ during the POL
and BG periods, respectively. Figure 3 illustrates the average PNSD and aerosol
chemical species in $PM_{2.5}$ during the two periods. The particle number concentrations
at all sizes were much higher during the POL period than during the BG period,



especially for ultrafine particles (with diameters, or $D_p$, < 100 nm). The diurnal
variation in PNSD during the POL period shown in Fig. 4a suggests that aerosol
particles with $D_p$ < 50 nm burst during rush hours and in the nighttime. The total particle
number concentration ($N$) remained greater than 30,000 cm$^{-3}$ at these times. However,
during the BG period, the number concentration of ultrafine particles only increased
weakly during rush hours or nucleation times. $N$ was always less than 20,000 cm$^{-3}$ on
all days during the BG period (Fig. 4b), probably linked with the reduction in on-road
vehicles during the holiday. As shown in Table 2, the ratio of BG to POL 10–50 nm
particle number concentrations ($N_{10-50\ nm}$) (0.47) is much smaller than the ratios for
larger particles (0.78 for $N_{50-100\ nm}$ and 0.67 for $N_{>100\ nm}$). These all demonstrate the
strong impact of reduced anthropogenic emissions on aerosol number concentrations,
especially for nucleation-mode and small Aitken-mode particles.

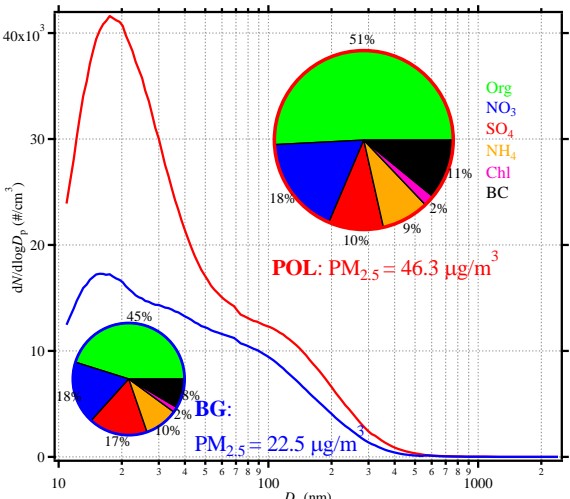


**Figure 3.** Average aerosol particle number size distributions (red and blue curves) and





mass fractions of aerosol chemical species in $PM_{2.5}$ (pie charts with red and blue
outlines) during the POL (in red) and BG (in blue) periods.

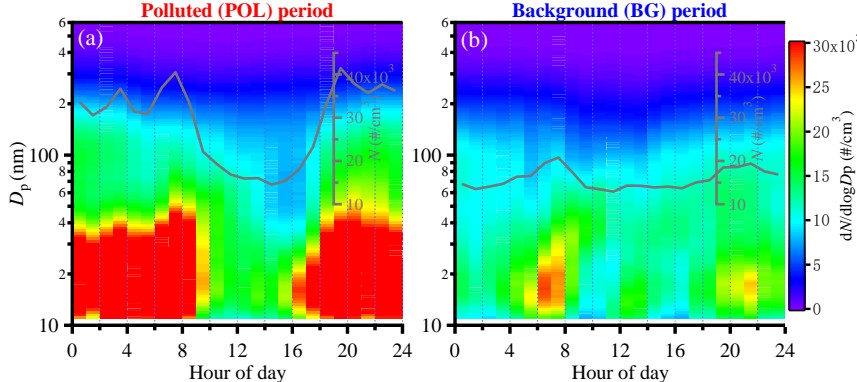


**Figure 4**. Diurnal variations in aerosol particle number size distribution (colored
background) and total aerosol number concentration ($N$, shown as grey curves) during
the (**a**) POL and (**b**) BG periods.

**Table 2**. Summary of the average aerosol number concentration ($N$) in different size
ranges, volume mixing ratios of trace gases, mass concentrations of $PM_{2.5}$ and different
aerosol chemical species, sulfur oxidation ratios (SOR), and nitrogen oxidation ratios
(NOR) during the POL and BG periods and their ratios.

| | $N_{10-50\,nm}$ (cm$^{-3}$) | $N_{50-100\,nm}$ (cm$^{-3}$) | $N_{>100\,nm}$ (cm$^{-3}$) | $SO_2$ (ppbv) | $NO_2$ (ppbv) | $O_3$ (ppbv) | $PM_{2.5}$ (μg/m$^3$) |
|---|---|---|---|---|---|---|---|
| POL | 20,861±19,935 | 3,946±2,544 | 3,888±2,757 | 8.31±6.35 | 51.96±27.35 | 26.06±22.24 | 46.32±39.05 |
| BG | 9,837±8,493 | 3,071±1,478 | 2,600±2,223 | 4.85±3.83 | 21.87±13.99 | 46.23±18.86 | 22.52±20.28 |
| BG/POL | 0.47 | 0.78 | 0.67 | 0.58 | 0.42 | 1.77 | 0.49 |

| | $m_{Org}$ (μg/m$^3$) | $m_{NO3}$ (μg/m$^3$) | $m_{SO4}$ (μg/m$^3$) | $m_{NH4}$ (μg/m$^3$) | $m_{BC}$ (μg/m$^3$) | SOR | NOR |
|---|---|---|---|---|---|---|---|
| POL | 23.55±19.58 | 8.25±7.91 | 4.59±6.20 | 3.96±3.83 | 5.05±4.51 | 0.27±0.17 | 0.09±0.08 |
| BG | 10.17±9.13 | 4.09±4.25 | 3.82±4.08 | 2.18±2.14 | 1.91±1.74 | 0.32±0.18 | 0.10±0.08 |
| BG/POL | 0.43 | 0.50 | 0.83 | 0.55 | 0.38 | 1.19 | 1.11 |


Table 2 also indicates that the mass concentrations ($m$) of aerosol chemical species
in $PM_{2.5}$ clearly decreased more during the BG period than during the POL period. The
$m$ related to primary emissions ($m_{org}$ and $m_{BC}$) decreased by more than 50 %, but the $m$
of secondary inorganic aerosols (SIA, including nitrate, sulfate, and ammonium)
slightly decreased, especially for sulfate ($m_{SO4}$ decreased by only 17 %). The pie charts
in Fig. 3 show significant differences in the aerosol chemical species of $PM_{2.5}$ during
the two periods. The mass fractions of Org and BC were lower during the BG period
(45 % and 8 %, respectively) than during the POL period (51 % and 11 %, respectively).
By contrast, the mass fraction of SIA was higher during the BG period (45 %) than
during the POL period (37 %). This indicates that the strongly reduced
anthropogenic emissions during the holiday caused sharp decreases in primary aerosols
but not secondary aerosols. The sulfur oxidation ratio (SOR) and nitrogen oxidation
ratio (NOR) are usually calculated to study the transformation of secondary aerosols
(Sun et al., 2006; Y. Li et al., 2017). SOR (NOR) is defined as the ratio of the molar
concentration of sulfate (nitrate) to the total molar concentration of sulfate (nitrate) and
$SO_2$ ($NO_2$). Table 2 shows that SOR and NOR were higher during the BG period than
during the POL period, suggesting that the formation of secondary inorganics was
enhanced during the BG period. Figure 5 shows that most large $PM_{2.5}$ mass
concentrations (> 100 $\mu g/m^3$) during the POL period occurred along with low RH (<
40 %) and low SIA mass fractions, indicating the important contribution of primary
emissions to the accumulation of $PM_{2.5}$ in a polluted environment. However, large
$PM_{2.5}$ mass concentrations (> 50 $\mu g/m^3$) during the BG period mainly appeared under



moderate RH (40 < RH < 80 %) and high SIA mass fraction conditions, likely caused
by enhanced aqueous-phase chemical reactions during this period.

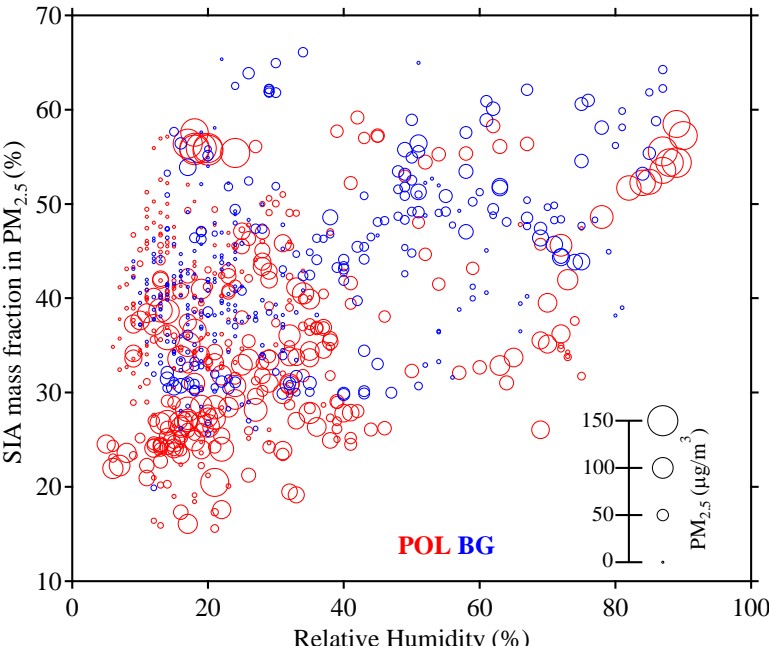


**Figure 5.** The variation in secondary inorganic aerosols (SIA) mass fraction in $PM_{2.5}$
as a function of ambient relative humidity during the POL (in red) and BG (in blue)
periods. The different circle sizes denote different $PM_{2.5}$ mass concentrations.

The diurnal variation in $[O_3]$ (Fig. 6a) shows more accumulated $O_3$ during the BG
period than during the POL period at any time of the day, revealing a stronger
atmospheric oxidation capacity during the BG period. In particular, $[O_3]$ at night was
two times higher during the BG period than during the POL period (Fig. 6b). Distinct
diurnal variations in SOR were found during the two periods (Fig. 6c). The higher SOR





at night during the BG period (Fig. 6c) indicates the enhanced transformation of $SO_2$ to
sulfate, likely related to nocturnal aqueous-phase chemical reactions. Figure S4
indicates that SOR increased following an increase in ALWC when ambient RH was
higher than ~40 % during the BG period. Moreover, Fig. 6d shows that the diurnal
variation in the SOR ratio during the two periods was similar to that of the $[O_3]$ ratio.
This suggests that sulfate formation during the holiday was likely enhanced by
nocturnal aqueous-phase chemical reactions between $SO_2$ and $O_3$. This is consistent
with the study of Fang et al. (2019), which found that ambient RH and the $O_3$
concentration are two prerequisites for rapid sulfate formation via aqueous-phase
oxidation reactions. This result highlights that controlling $O_3$ formation under current
emission conditions in winter in Beijing is key to further reducing the formation of
sulfate and implies that the high underestimation of sulfate at night in models (Miao et
al., 2020) could be caused by the inaccurate simulation of $[O_3]$. The higher daytime
NOR (Fig. 6e) than nighttime NOR during the two periods illustrates that the formation
of nitrate was mainly controlled by photochemical reactions in winter. Figure 6f shows
that the larger NOR difference (the higher NOR ratio) during rush hours during the two
periods likely occurred because a mass of emitted $NO_x$ during rush hours could not be
transformed to nitrate during the POL period. Figure 6e and 6f also suggests nitrate
formation was enhanced a little during the holiday likely due to the enhanced daytime
photochemical reactions.

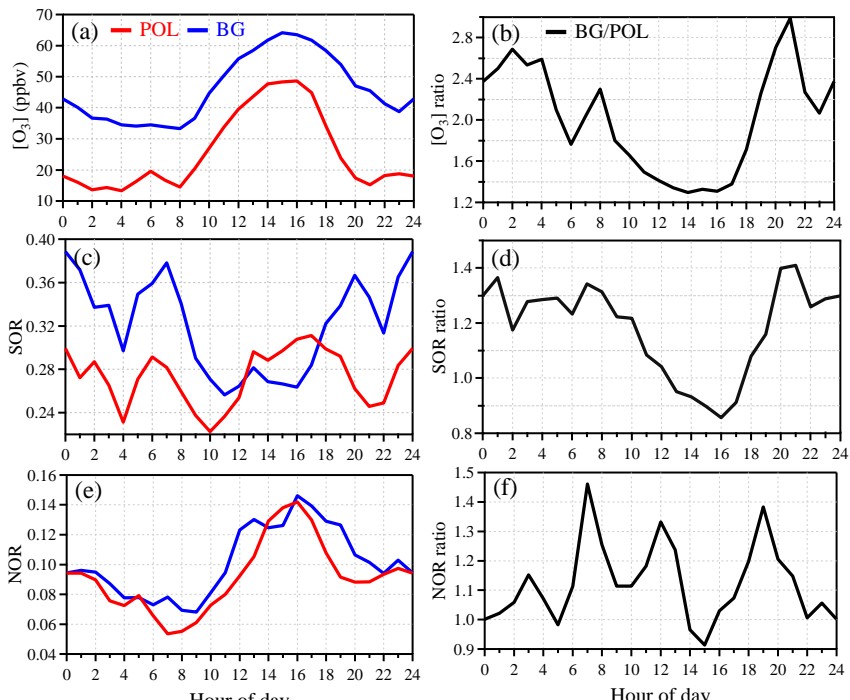


**Figure 6.** Diurnal variations in (**a** and **b**) $O_3$ volume mixing ratio and its ratio, (**c** and

**d**) sulfur oxidation ratio (SOR) and its ratio, and (**e** and **f**) nitrogen oxidation ratio

(NOR) and its ratio during the BG and POL periods. The ratio of a quantity is that

quantity during the BG period divided by that quantity during the POL period.

Overall, the reduced anthropogenic emissions led to a drastic decrease in aerosol

particle number concentration during the holiday. However, the atmospheric oxidation

capacity was enhanced during the holiday, thereby promoting the formation of

secondary inorganics (especially sulfate).



**3.3.** **Impact of reduced anthropogenic emissions on regional air pollution**


Figure 7a-c shows that the volume mixing ratios of emitted trace gases ([SO$_2$],
[NO$_2$], and [CO]) decreased in the BTH region during the BG period. [NO$_2$] and
[SO$_2$] decreased by more than 40 % in all cities and 35 % in heavy-industry cities
(distributed in the southern and northeastern parts of the BTH region). This indicates a
regional reduction in anthropogenic emissions during the holiday. Figure 7d shows
that [O$_3$] increased in all cities and that the increase was more than 50 % in all cities
except for a high-altitude city (Zhangjiakou) to the northwest, implying the regional
enhancement of the atmospheric oxidation capacity.

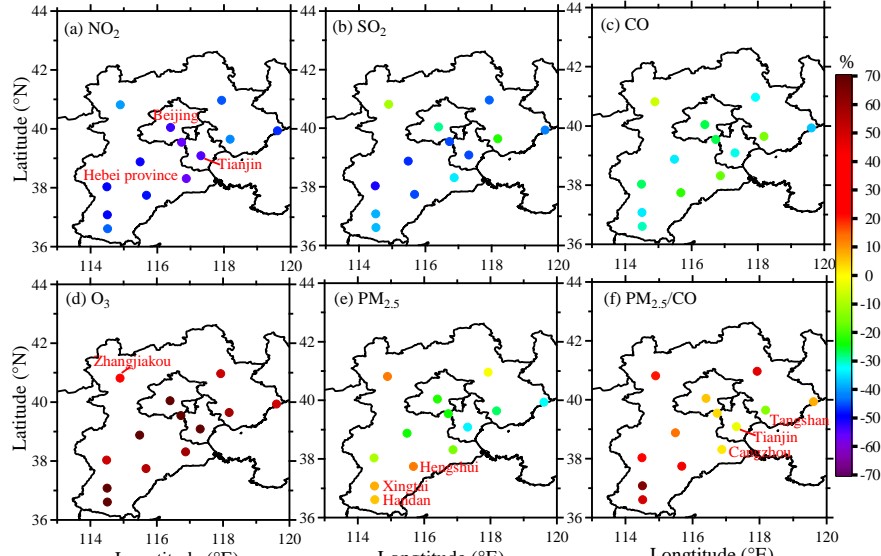


**Figure 7**. Percent changes in trace gas volume mixing ratios (NO$_2$, SO$_2$, CO, and O$_3$),
the PM$_{2.5}$ mass concentration, and the PM$_{2.5}$/CO ratio during the BG period relative to
the POL period: $100 \times \left( \frac{[BG]-[POL]}{[POL]} \right)$.



The $PM_{2.5}$ mass concentration decreased at a much lower rate relative to the
decrease in $[NO_2]$ and $[SO_2]$ in most cities during the BG period (Fig. 7e), while it
increased slightly at Zhangjiakou and three southern heavy-industry cities (Hengshui,
Xingtai, and Handan). The ratio $PM_{2.5}$/CO is an indicator of aerosol secondary
formation to primary emissions. An increase in $PM_{2.5}$/CO was found during the BG
period at all cities except for three coastal cities (Tangshan, Tianjin, and Cangzhou),
revealing the regional enhancement of secondary aerosol formation during the
holiday. The weak decrease in $PM_{2.5}$/CO at the three coastal cities was likely due to
the influence of mixed sea flows.
The regional analysis of air pollution assumes that the findings from Beijing
presented in section 3.2 are applicable to the entire BTH region. Regionally reduced
anthropogenic emissions resulted in sharply decreased gaseous pollutants and
increased $O_3$. Higher atmospheric oxidation led to the enhanced formation of
secondary aerosols, thus counteracting the decrease in $PM_{2.5}$ mass concentration.
There are two possible reasons explaining the high $[O_3]$ during the holiday: (1) the
reduced gaseous precursors ($NO_x$ and $SO_2$) weakened the consumption of $O_3$, and (2)
$O_3$ formation in the BTH region is volatile organic compound (VOC)-controlled under
current emission conditions, therefore the reduction in $NO_x$ would lead to higher $[O_3]$.
This result demonstrates that it is more important to reduce VOC emissions to control
$PM_{2.5}$ in winter in the BTH region.



### 4. Conclusions and Implications

In recent years, the mass concentration of particulate matter with an aerodynamic diameter of less than 2.5 μm ($PM_{2.5}$) has shown a general decreasing trend, presumably due to the series of emission reduction measures taken in China attempting to improve air quality. However, haze pollution episodes still occur from time to time, including during some special events when primary emissions reduced drastically, such as the Chinese New Year holiday and even during the COVID-19 lockdown when anthropogenic activities diminished drastically (X. Huang et al., 2020). We conjecture that reductions through primary emissions may be offset by increases in the formation of secondary aerosols.

To test this, we examined the secondary aerosol formation mechanism in a comprehensive field experiment conducted in Beijing. Comprehensive aerosol and meteorological measurements were made for more than two years, but data around the 2019 Chinese Spring Festival from 16 January to 17 February were employed in this study to single out the impact of emission reductions due to the holiday. The study period was divided into polluted (POL) and background (BG) periods, with high and low anthropogenic emissions before and during the festival holiday, respectively. Investigated were the impacts of reduced anthropogenic emissions on trace gases and $PM_{2.5}$ under similar meteorological conditions.

The average $PM_{2.5}$ mass concentrations were 46.3 and 22.5 μg/m$^3$ during the POL and BG periods, respectively, with no heavy haze events occurring. The average aerosol particle number size distribution shows that the reduced anthropogenic emissions



during the holiday led to decreased aerosol number concentrations at all sizes,
especially in the nucleation and Aitken modes (mobility diameters less than 50 nm).
Simultaneously, the reduced anthropogenic emissions resulted in decreases in the
volume mixing ratios of $SO_2$ and $NO_2$ and an unexpected increase in the volume mixing
ratio of $O_3$ [$O_3$] during the BG period. The analysis of the aerosol chemical species in
$PM_{2.5}$ demonstrates that the large decreases in organics and black carbon mass
concentrations during the BG period were likely caused by the large decrease in on-
road vehicles. Moreover, the mass concentration of nitrate also decreased while that of
sulfate decreased much less during the BG period. Comparisons of the sulfur oxidation
ratio (SOR) and the nitrogen oxidation ratio (NOR) during the two periods imply that
the transformation of gaseous precursors to secondary inorganics (especially the
transformation of $SO_2$ to sulfate) was promoted during the BG period, likely due to the
enhanced atmospheric oxidation capacity. The diurnal variation in the SOR ratio
between the BG and POL periods was similar to that of the [$O_3$] ratio, illustrating that
sulfate formation was promoted by the enhanced nocturnal aqueous-phase chemical
reactions between $SO_2$ and $O_3$ under moderate relative humidity (RH) conditions (40 %
< RH < 80 %). The higher NOR in the daytime during the two periods points out that
the formation of nitrate was mainly controlled by photochemical reactions and weakly
affected by the increase in [$O_3$].
This study also investigated the impact of reduced anthropogenic emissions on
regional air pollution patterns during the holiday. The variation trends of trace gases in
most cities in the Beijing-Tian-Hebei (BTH) region were similar to those in Beijing,



indicating the regional influence of reduced anthropogenic emissions on the volume
mixing ratios of trace gases during the holiday. The weak $PM_{2.5}$ variation and the
increased $PM_{2.5}/CO$ ratio (an indicator of aerosol secondary formation to primary
emissions) during the BG period both suggest that the enhanced formation of secondary
aerosols offset the regional decrease in $PM_{2.5}$ during the holiday.

Our findings provide evidence that decreases in anthropogenic emissions can

promote the formation of secondary inorganics due to the enhancement of the
atmospheric oxidation capacity (manifested by more accumulated $O_3$). In the future, the
simultaneous control of $PM_{2.5}$ and $O_3$ will be needed to further reduce air pollution. The
$O_3$ formation in winter in the BTH region is possibly volatile organic compound
(VOC)-controlled under current emission conditions. Controlling VOC emissions may
thus be more important than controlling emissions of nitrogen oxides.

*Acknowledgement.* This work was funded by the National Key R&D Program of the
Ministry of Science and Technology, China (Grant No. 2017YFC1501702) and the
Startup Foundation for Introducing Talent of NUIST (No. 2019r077).

*Data availability.* Data from the Chinese Ministry of Ecology and Environment
network and Beijing Municipal Environmental Monitoring Center can be downloaded
from the websites given in the main text. The measurement data from the field
experiment used in this study are available from the first author upon request
(yuyingwang@nuist.edu.cn).




*Author contributions.* ZL and PY designed the field experiment. YW and ZL

conceived the study and led the overall scientific questions. YW, QW, and XJ

processed the measurement data and prepared this paper. MC copyedited the article.

Other co-authors participated in the implementation of this experiment and the

discussion of this paper.

*Competing interests.* The authors declare that they have no conflict of interest.

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
