# Peer review of "Enhancement of secondary aerosol formation by reduced anthropogenic emissions during Spring Festival 2019 and enlightenment for regional PM2.5 control in Beijing"

_Atmospheric Chemistry and Physics, 2020_

## Referee Comment (RC1) · Anonymous Referee #1 · 19 Jul 2020

This work present the results of the field measurement of air pollution during 2019 Spring Festival. Spring Festival is a special period to investigate the impact of emission reduction on air quality. The topic itself is very interesting. The authors provide very interesting data. However, there's major defect for the current manuscript. The authors compared the variation of various air pollutants, and gave the conclusion that the reduction of "Sharply reduced sulfur dioxide (SO2) and nitrogen dioxide (NO2) concentrations during the festival holiday resulted in an unexpected increase in the surface ozone (O3) concentration", and further promote the secondary formation. These con-

clusions are astonishing and new, but the authors did not provide enough convincing evidence. Besides, considering the quality of ACP, I will not recommand the publication of this manuscript.
* * *

---

## Referee Comment (RC2) · Anonymous Referee #2 · 12 Oct 2020

Comments on manuscript entitled "Enhancement of secondary aerosol formation by reduced anthropogenic emissions during Spring Festival 2019 and enlightenment for regional PM2.5 control in Beijing"

General comments: This manuscript reported primary pollutant reduction but enhanced SIA formation in an emission reduction period during the 2019 Spring Festival in Beijing. The opposite trend of atmospheric oxidative capacity responding emission reduction was proposed the cause for enhanced SIA formation. Though the supporting discussion still appeared to be weak. Nevertheless, this study should call for the attention on SIA pollution control policy mitigation. I thus recommend publication of this manuscript on ACP with minor revision.

Specific comments: Lines 21-22: O3 control regime on a regional scale is still a controversial topic. This manuscript did not intend to discuss on such topic given no VOCs measurements were present. Therefore, it is a bit risky to go such far with current data available. I suggest to delete the statement on NOx and VOCs control strategy if no more discussion shall add. Line 56: be specific! Change to "secondary inorganic aerosol formation" Line 92: consider to revise this sentence Line 160: as shown in Fig.2, O3 titration appeared to occur in both POL and BG period. Ox=O3+NO2 is thus suggested to add in Fig.2. Lines 202-203: cannot read from Figure 2 that morg and mBC increase by % at night from daytime is less in BG relative to POL Lines 203-204: both mnitrate and msulfate varied! Lines 246-24: decreased from what? Lines 255-257: From the context, I can only get that Org and BC reduction was sharper than sulfate and nitrate. If I can accept that "secondary (inorganic) aerosol" could replace "sulfate and nitrate", I am still reluctant to accept that Org and BC are all primary aerosol. Figure 5: The high SIA and large PM2.5 number in POL were mostly seen at low RH, which is against the impression that heavy PM2.5 pollution was usually accompanied by high RH condition in literature. The author should at least address such unusual data. Figure 6: Given the discussion on RH or ALWC in the context, I would suggest to add one of the two parameters in one column. Line 403: High O3 concentration itself will not surely lead to strong atmospheric oxidative capacity or even O3 production. The first reason is that O3 was titrated in Figure 2. The secondary reason is that O3 can be regionally transported as a relatively long-lived species. And the third, OH instead of O3 is the major oxidant in the atmosphere, which better represents the atmospheric oxidative capacity and does not differ significantly from pollution days to clean days in winter Beijing (see Eloise et al., Elevated levels of OH observed in haze events during wintertime in central Beijign). More data or discussion are needed here.

---

## Author Comment (AC1) · 28 Oct 2020

This work present the results of the field measurement of air pollution during 2019 Spring Festival. Spring Festival is a special period to investigate the impact of emission reduction on air quality. The topic itself is very interesting. The authors provide very interesting data. However, there's major defect for the current manuscript. The authors compared the variation of various air pollutants, and gave the conclusion that the reduction of "Sharply reduced sulfur dioxide (SO2) and nitrogen dioxide (NO2) concentrations during the festival holiday resulted in an unexpected increase in the surface ozone (O3) concentration", and further promote the secondary formation. These conclusions are astonishing and new, but the authors did not provide enough convincing evidence. Besides, considering the quality of ACP, I will not recommand the publication of this manuscript.

**Response:** While we appreciate the critical comment of the review, it'd be much more helpful if the reviewer could have provided a more informative and insightful comment so that we know more about his/her concern. For any scientific research, a finding of "astonishing and new" should not be the reason for rejection.

In this study, we investigated the impact of emission reductions on the concentrations of several trace gases and their further impact on aerosol formation during the special period of the 2019 Spring Festival. It is clear that emission reductions could efficiently reduce primary pollutants ($SO_2$, $NO_x$, BC, etc.). The time series of $O_x$ ($O_3$+$NO_2$) added in Fig. 2 depicts a weak decrease of $O_x$ from the POL period to the BG period, suggesting that the possible appeared $O_3$-titration made [$O_3$] increase during the BG period. Simultaneously, the mass concentrations of secondary inorganics decreased but their reduction percentages were much lower than those of primary pollutants. With the further analysis of SOR, NOR, and their relationships with ambient RH and ALWC, we concluded that the enhancement of aqueous chemical reactions oxidized by the dissolved $O_3$ maybe the main reason causing the enhanced secondary inorganic aerosol (SIA) formation, especially for sulfate.

In recent years, the annual average $PM_{2.5}$ concentration has decreased rapidly in China, benefitting from the implementation of many emission reduction measures taken by the Chinese government. However, the mass fraction of inorganics increased by more than 10 % during these years (H. Li et al., 2019; Y. Wang et al., 2019), implying the formation enhancement of secondary inorganic aerosols (SIA), which partly counteracted the decrease in $PM_{2.5}$. Xie et al. (2020) found that the aerosol pH level increased as $PM_{2.5}$ decreased in urban Beijing because of the increased mass ratio of nitrate to sulfate. They also stated that the major chemical processes during haze events and the control target should be re-evaluated to obtain the most effective control strategy. As one possible consequence of the increased aerosol pH, the dissolved $O_3$ in particles may play a more important role in SIA formation, especially for sulfate (Seinfeld and Pandis, 2016). Our study provides sound evidence for this.

Other recent studies have also suggested that the role of $O_3$ on SIA formation cannot be neglected. For example, Fang et al. (2019) found that relative humidity (RH) and $O_3$ concentration were two important prerequisites for sulfate formation, based on a year-long set of field measurements made in Beijing. They found a rapid rise in the SOR at the RH

threshold of ~ 45% or an $O_3$ concentration threshold of ~35 ppb, similar to what we found in our study. As another example, Huang et al. (2020) investigated air quality during the COVID-19 lockdown using comprehensive measurements and modeling with a focus on China. They also found that a large reduction in emissions could enhance the concentration of O3 in winter in the Beijing-Tianjin-Hebei (BTH) region, promoting SIA formation through the enhancement of nocturnal aqueous chemical reactions during the COVID-19 lockdown.

**References**

Fang, Y., Ye, C., Wang, J., Wu, Y., Hu, M., Lin, W., Xu, F., and Zhu, T.: Relative humidity and O3 concentration as two prerequisites for sulfate formation, Atmos. Chem. Phys., 19, 12,295-12,307, https://doi.org/10.5194/acp-19-12295-2019, 2019.

Huang, X., Ding, A., Gao, J., Zheng, B., Zhou, D., Qi, X., Tang, R., Wang, J., Ren, C., Nie, W., Chi, X., Xu, Z., Chen, L., Li, Y., Che, F., Pang, N., Wang, H., Tong, D., Qin, W., Cheng, W., Liu, W., Fu, Q., Liu, B., Chai, F., Davis, S. J., Zhang, Q., and He, K.: Enhanced secondary pollution offset reduction of primary emissions during COVID-19 lockdown in China, National Science Review, https://doi.org/10.1093/nsr/nwaa137, 2020.

Li, H., Cheng, J., Zhang, Q., Zheng, B., Zhang, Y., Zheng, G., and He, K.: Rapid transition in winter aerosol composition in Beijing from 2014 to 2017: response to clean air actions, Atmos. Chem. Phys., 19, 11,485–11,499, https://doi.org/10.5194/acp-19-11485-2019, 2019.

Seinfeld, J. H., and Pandis, S. N.: Atmospheric chemistry and physics: from air pollution to climate change, edited, John Wiley & Sons, 2016.

Wang, Y., Chen, J., Wang, Q., Qin, Q., Ye, J., Han, Y., Li, L., Zhen, W., Zhi, Q., Zhang, Y., and Cao, J.: Increased secondary aerosol contribution and possible processing on polluted winter days in China, Environ. Int., 127, 78–84, https://doi.org/10.1016/j.envint.2019.03.021, 2019.

Xie, Y., Wang, G., Wang, X., Chen, J., Chen, Y., Tang, G., Wang, L., Ge, S., Xue, G., Wang, Y., and Gao, J.: Nitrate-dominated PM2.5 and elevation of particle pH observed in urban Beijing during the winter of 2017, Atmos. Chem. Phys., 20, 5019-5033, https://doi.org/10.5194/acp-20-5019-2020, 2020.

---

## Author Comment (AC2) · 28 Oct 2020

Comments on manuscript entitled "Enhancement of secondary aerosol formation by reduced anthropogenic emissions during Spring Festival 2019 and enlightenment for regional PM2.5 control in Beijing"

General comments: This manuscript reported primary pollutant reduction but enhanced SIA formation in an emission reduction period during the 2019 Spring Festival in Beijing. The opposite trend of atmospheric oxidative capacity responding emission reduction was proposed the cause for enhanced SIA formation. Though the supporting discussion still appeared to be weak. Nevertheless, this study should call for the attention on SIA pollution control policy mitigation. I thus recommend publication of this manuscript on ACP with minor revision.

Specific comments:

Lines 21-22: O3 control regime on a regional scale is still a controversial topic. This manuscript did not intend to discuss on such topic given no VOCs measurements were present. Therefore, it is a bit risky to go such far with current data available. I suggest to delete the statement on NOx and VOCs control strategy if no more discussion shall add.

**Response:** Agreed. The sentence "The emission control of volatile organic compounds (VOCs) may be more suitable than the emission control of $NO_x$ to reduce $O_3$ because VOCs under current emission conditions likely control the formation of $O_3$ in winter in the BTH region" has been deleted.

Line 56: be specific! Change to "secondary inorganic aerosol formation"

**Response:** Revised.

Line 92: consider to revise this sentence

**Response:** This sentence has been revised as: "Our measurements around this period of the field campaign are thus ideal for investigating the impact of reduced anthropogenic emissions on surface $O_3$ and aerosol formation.".

Line 160: as shown in Fig.2, O3 titration appeared to occur in both POL and BG period. Ox=O3+NO2 is thus suggested to add in Fig.2.

**Response:** The time series of $O_x$ ($O_3+NO_2$) has been added to Fig. 2 in the manuscript (shown below as Fig. R1). It shows a weak variation of $O_x$ from the POL period to the BG period, indicating that the presence of strong $O_3$-titration during Spring Festival 2019. The corresponding discussion about $O_x$ and $O_3$ titration has been added to section 3.1 of the revised manuscript.

[Figure]

Figure R1. Time series of (a) ambient temperature (T) and relative humidity (RH), (b) wind direction (WD) and speed (WS), (c) volume mixing ratios of trace gases [$O_3$, $SO_2$, $NO_2$ and $O_x$ ($O_3+NO_2$)], (d) the aerosol particle number size distribution measured by the SMPS, and (e) mass concentrations of aerosol chemical species in $PM_{2.5}$ measured by the ACSM and the AE-33. The trace gas information was from the Yizhuang station, and the others were observed at the experiment site in Beijing (16 January to 17 February 2019).

Lines 202-203: cannot read from Figure 2 that morg and mBC increase by % at night from daytime is less in BG relative to POL Lines

**Response:** We're sorry that this sentence has confused the reviewer. Figure 2e depicts that the peaks of $m_{org}$ and $m_{BC}$ at night during the BG period were much lower than those during the POL period, caused by emission reductions during the BG period. Therefore, here we want to express that the enhancement of $m_{org}$ and $m_{BC}$ at night during the BG period was not as strong as that during the POL period.

This sentence has been revised as: "However, the increases in $m_{org}$ and $m_{BC}$ at night during the BG period were not as strong as those during the POL period.".

203-204: both mnitrate and msulfate varied!

**Response:** Figure 2e depicts that both $m_{NO3}$ and $m_{SO4}$ decreased from the POL period to the POL period. However, their reduction magnitudes differed considerably.

Lines 246-24: decreased from what?

**Response:** This sentence has been revised as:" Table 2 also indicates that the mass concentrations ($m$) of aerosol chemical species in $PM_{2.5}$ were much less during the BG period than during the POL period".

Lines 255-257: From the context, I can only get that Org and BC reduction was sharper than sulfate and nitrate. If I can accept that "secondary (inorganic) aerosol" could replace "sulfate and nitrate", I am still reluctant to accept that Org and BC are all primary aerosol.

**Response:** The reviewer asks a good question. BC is mainly from primary emissions, but organics were not. Part of the organics is from primary emissions (i.e., primary organic aerosols, or POA), but another part is from gas-to-particle transformations (i.e., secondary organic aerosols, or SOA). Unfortunately, we are not able to separate POA and SOA in organics using our measurement data from the campaign. In this paper, we were not trying to define BC and organics as the primary matter. They were simply regarded as representing primary matter because many of them were from primary emissions. To a certain extent, the mass variations of BC and organics can represent the mass variations of primary aerosols. Similarly, SIA matter (mainly sulfate, nitrate and ammonium) are important chemical components of secondary aerosols, so their mass variations can represent the mass variations of secondary aerosols.

Figure 5: The high SIA and large PM2.5 number in POL were mostly seen at low RH, which is against the impression that heavy PM2.5 pollution was usually accompanied by high RH condition in literature. The author should at least address such unusual data.

**Response:** Some studies have found that heavy haze events are generally associated with high RH conditions and southerly winds. This is because the southerly winds are not only beneficial to the transport of pollutants from southern highly industrialized areas, but also to the transport of water vapor. In our study, the prevailing winds during both the POL and BG periods were northerly, which were beneficial to dispersing pollutants in Beijing, so no heavy haze episodes occurred during the two periods. However, the $PM_{2.5}$ during the POL period with ordinary emission conditions could reach moderate pollution level (over 100 $\mu g/m^3$) although the ambient RH was low.

    The basic meteorological and environmental characteristics have been described in section 3.1.

Figure 6: Given the discussion on RH or ALWC in the context, I would suggest to add one of the two parameters in one column.

**Response:** That is a good suggestion. A figure showing the diurnal variation in ambient relative humidity (RH) (Fig. R2 below) was added to the supplement (Fig. S4). It shows that the ambient RH levels at night are elevated during both the POL and BG periods, favorable for aqueous chemical reactions.

[Figure]

Figure R2. Diurnal variation of ambient relative humidity (RH) during the POL and BG periods.

Line 403: High O3 concentration itself will not surely lead to strong atmospheric oxidative capacity or even O3 production. The first reason is that O3 was titrated in Figure 2. The secondary reason is that O3 can be regionally transported as a relatively long-lived species. And the third, OH instead of O3 is the major oxidant in the atmosphere, which better represents the atmospheric oxidative capacity and does not differ significantly from pollution days to clean days in winter Beijing (see Eloise et al., Elevated levels of OH observed in haze events during wintertime in central Beijign). More data or discussion are needed here.

**Response:** Agreed. The analysis of $O_x$ above shows that $O_3$-titration appeared during the special period studied. In this campaign, OH was not measured, so the atmospheric oxidation capacity wasn't analyzed accurately. For this reason, the discussion about atmospheric oxidation capacity in the manuscript has been deleted.